# Profiling the Abiotic Stress Responsive microRNA Landscape of *Arabidopsis thaliana*

**DOI:** 10.3390/plants8030058

**Published:** 2019-03-10

**Authors:** Joseph L. Pegler, Jackson M. J. Oultram, Christopher P. L. Grof, Andrew L. Eamens

**Affiliations:** Centre for Plant Science, School of Environmental and Life Sciences, Faculty of Science, University of Newcastle, Callaghan 2308, Australia; Joseph.Pegler@uon.edu.au (J.L.P.); Jackson.Oultram@uon.edu.au (J.M.J.O.); chris.grof@newcastle.edu.au (C.P.L.G.)

**Keywords:** *Arabidopsis thaliana*, abiotic stress, heat stress, drought stress, salt stress, microRNAs (miRNAs), miRNA target gene expression, RT-qPCR

## Abstract

It is well established among interdisciplinary researchers that there is an urgent need to address the negative impacts that accompany climate change. One such negative impact is the increased prevalence of unfavorable environmental conditions that significantly contribute to reduced agricultural yield. Plant microRNAs (miRNAs) are key gene expression regulators that control development, defense against invading pathogens and adaptation to abiotic stress. *Arabidopsis thaliana* (*Arabidopsis*) can be readily molecularly manipulated, therefore offering an excellent experimental system to alter the profile of abiotic stress responsive miRNA/target gene expression modules to determine whether such modification enables *Arabidopsis* to express an altered abiotic stress response phenotype. Towards this goal, high throughput sequencing was used to profile the miRNA landscape of *Arabidopsis* whole seedlings exposed to heat, drought and salt stress, and identified 121, 123 and 118 miRNAs with a greater than 2-fold altered abundance, respectively. Quantitative reverse transcriptase polymerase chain reaction (RT-qPCR) was next employed to experimentally validate miRNA abundance fold changes, and to document reciprocal expression trends for the target genes of miRNAs determined abiotic stress responsive. RT-qPCR also demonstrated that each miRNA/target gene expression module determined to be abiotic stress responsive in *Arabidopsis* whole seedlings was reflective of altered miRNA/target gene abundance in *Arabidopsis* root and shoot tissues post salt stress exposure. Taken together, the data presented here offers an excellent starting platform to identify the miRNA/target gene expression modules for future molecular manipulation to generate plant lines that display an altered response phenotype to abiotic stress.

## 1. Introduction

Anthropogenically driven climate change is a rapidly growing concern globally, forcing interdisciplinary research collaborations to provide solutions that address and/or negate the numerous negative consequences of a changing climate, with the provision of sustainable food security the overarching goal of contemporary agricultural research [1,2,3]. Throughout the last half-century, agriculture has attempted to continue to achieve the food demands of an ever-growing global population via the unsustainable practice of clearing biodiverse terrestrial ecosystems for additional cultivation of traditional cropping species, an alarming practice that further contributes to the global carbon footprint and climate change [4,5]. Considering the capability limitation of the current maximum annual global crop yield to land area ratio, it is obvious that alternate strategies are now required if cropping agriculture is to continue to ensure food security, whilst terminating unsustainable farming practices, and while achieving these goals in an ever increasingly unfavorable environment [4].

Lacking the mobility of metazoa, the sessile nature of a plant requires intricate and interrelated gene expression networks to mediate the plant’s ability to physiologically and phenotypically respond to its surrounding environment. Such multilayered molecular networks are especially important to a plant’s adaptive and/or defensive response when either abiotic or biotic stress is encountered [6,7]. Elucidating the gene expression cascades that underpin the ability of a plant to adapt to, or mitigate, the negative impact of abiotic stress is the first key step in the development of new plant lines harboring molecular modifications which allow the plant to display an altered response phenotype when exposed to abiotic stress. The genetic model plant, *Arabidopsis thaliana* (*Arabidopsis*), is readily amenable to molecular modification, thereby offering plant biology researchers an excellent experimental system to validate which introduced molecular modifications mediate the expression of abiotic stress tolerance phenotypes.

Since their initial identification in *Arabidopsis* in 2002 [8], plant microRNAs (miRNAs), small non-protein-coding regulatory RNAs, have been repeatedly demonstrated to be key regulators of gene expression across all phases of plant development [9,10], in mediating a defense response against invading viral, bacterial or fungal pathogens [11,12], or to direct a plant’s adaptive response to exposure to abiotic stress, including the stresses of heat, drought and salt stress [13,14]. Each *Arabidopsis* miRNA is processed from a stem-loop structured precursor transcript, a non-protein-coding RNA that has folded back upon itself to form this structure, post RNA polymerase II (Pol II)-catalyzed transcription from a unique *MICRORNA* (*MIR*) gene [15,16,17]. Like protein coding loci, the promoter regions of many *MIR* genes harbor *cis*-elements that contribute to the control of *MIR* gene expression in response to numerous signals external to the cell, including the signals that stem from abiotic stress [18,19,20]. Altered *MIR* gene expression, and therefore altered mature miRNA abundance, in turn leads to changes in miRNA target gene expression, with each miRNA loaded by the miRNA-induced silencing complex (miRISC) to be used as a sequence specificity guide to modulate target gene expression via either a messenger RNA (mRNA) cleavage or translational repression mechanism of RNA silencing [9,21,22,23]. To date, in *Arabidopsis*, numerous miRNA/target gene expression modules have been demonstrated to be responsive to abiotic stress, with alteration to the molecular profile of some expression modules further shown to assist the plant to adapt to abiotic stress due to molecular-driven changes to key pathways, such as the photosynthesis, sugar signaling, stomatal control and hormone signaling pathways [6,7]. Of particular interest is the demonstration that considerable numbers of *MIR* gene families identified as abiotic stress responsive in *Arabidopsis*, play a conserved functional role across phylogenetically diverse dicotyledonous and monocotyledonous species, including many of the major monocot grasses (such as *Zea mays*, *Oryza sativa* and *Triticum aestivum*) cultivated to provide much of the daily calorific intake of the world’s population [24,25,26,27,28]. This demonstration identifies the use of *Arabidopsis* as an ideal experimental system to molecularly modify the profile of such conserved abiotic stress responsive miRNA expression modules to determine if the introduced modifications enable *Arabidopsis* to display an altered response phenotype to abiotic stress.

High throughput sequencing was therefore employed here to profile the miRNA landscape of wild-type *Arabidopsis* plants exposed to heat, drought and salt stress. Sequencing identified large miRNA cohorts responsive to each applied stress, with 121, 123 and 118 miRNA sRNAs determined to have a greater than 2.0-fold abundance change post heat, drought and salt stress treatment of *Arabidopsis* whole seedlings, respectively. For each assessed stress, a quantitative reverse transcriptase polymerase chain reaction (RT-qPCR)-based approach was used to experimentally confirm the abundance of five miRNA sRNAs. For each miRNA experimentally validated to have altered abundance post stress exposure, RT-qPCR was additionally used to document reciprocal target gene expression profiles to further confirm each miRNA/target gene expression module as abiotic stress responsive. Sequencing, and the initial RT-qPCR profiling of miRNA abundance and miRNA target gene expression was performed on whole seedling samples, therefore RT-qPCR was next employed to confirm the documented whole seedling expression trends in *Arabidopsis* root and shoot tissues post salt stress exposure. This analysis showed that the initial abiotic stress responsive miRNA expression profiles identified in whole plant samples were an accurate representation of the tissue-specific profile of each assessed expression module. Taken together, the data presented here identifies numerous miRNA/target gene expression modules that could be targeted for future molecular modification to determine if such modification allows *Arabidopsis* to display an altered response to abiotic stress. Furthermore, the information gathered in *Arabidopsis* using such a molecular approach could be potentially translated to an agronomically important cropping species for the future generation of plant lines that display an adaptive phenotypic response to abiotic stress.

## 2. Results

### 2.1. Response of Wild-type Arabidopsis Seedlings to Heat, Drought and Salt Stress

Post germination and cultivation on standard growth media, 8 day old wild-type *Arabidopsis* seedlings (ecotype Columbia-0 (Col-0)) were exposed to a 7-day period of either heat, drought or salt stress. Figure 1A displays the phenotypes expressed by heat, drought and salt stressed Col-0 plants, compared to that of 15 day old, non-stressed wild-type *Arabidopsis*. The growth of drought (mannitol supplemented media) and salt stressed plants was significantly repressed, to differing degrees, after 7 days of exposure to both stress treatments, as readily demonstrated by a reduction to rosette tissue fresh weight (Figure 1B), and rosette diameter (Figure 1C). Furthermore, salt stress treatment induced the accumulation of anthocyanin (Figure 1D), an antioxidant produced by plants to combat the cellular stress caused by reactive oxygen species [29,30], specifically in the shoot apex of salt stressed *Arabidopsis* whole seedlings (Figure 1A). Unlike the reductions observed to rosette tissue fresh weight and the diameter of the rosette of drought and salt stressed Col-0 plants, the 7-day heat stress treatment resulted in the promotion of both of these phenotypic parameters. Specifically, compared to non-stressed control plants, the rosette leaf fresh weight (Figure 1B), and rosette diameter (Figure 1C), were increased by 75% and 127% respectively, in heat stressed Col-0 plants. Promotion of specific aerial tissue growth parameters, namely hypocotyl and rosette leaf petiole elongation, has been reported previously for *Arabidopsis* plants cultivated under conditions of elevated temperature [31,32]. Although promotion of aerial tissue growth suggested that the 7-day heat stress treatment had a positive influence on *Arabidopsis* development, the 177% increase in anthocyanin accumulation observed in parallel (Figure 1D), alternatively suggested that this treatment actually induced high levels of stress in *Arabidopsis* cells, specifically the cells of the shoot apex and rosette leaf petioles (Figure 1A). Therefore, to additionally demonstrate that each applied stress was influencing *Arabidopsis* at the molecular level, the expression of the well characterized stress-induced gene, *Δ1-PYRROLINE-5-CARBOXYLATE SYNTHETASE1* (*P5CS1*; *AT2G39800*) [33,34,35] was quantified by RT-qPCR. This analysis revealed that *P5CS1* transcript abundance was upregulated 3.0-, 1.7- and 45-fold in heat, drought and salt stressed *Arabidopsis* whole seedlings, respectively (Figure 1E).

### 2.2. Profiling of the microRNA Landscape of Heat, Drought and Salt Stressed Arabidopsis Whole Seedlings

Demonstrated induction of *P5CS1* expression (Figure 1E), a well characterized [33,34,35] stress-induced gene in *Arabidopsis*, post exposure to heat, drought and salt stress, suggested that all three applied stresses were inducing molecular responses in *Arabidopsis*. Therefore, total RNA was extracted from non-stressed plants, and from heat, drought and salt stressed *Arabidopsis* whole seedlings, and the sRNA fraction of each analyzed via high throughput sequencing to profile the respective miRNA landscapes (Figure 2A). In total, 333 miRNA sRNAs were identified by sequencing across the control and stress treatments (see Appendix A). Sequencing further revealed a greater than 2-fold abundance change for 121, 123 and 118 mature miRNA sRNAs for heat, drought and salt stressed *Arabidopsis* whole seedlings, respectively (Figure 2B). More specifically, heat stress promoted the accumulation of 90 miRNAs (miR395a abundance was increased to the greatest degree at 89.4-fold) and repressed the abundance of 17 miRNAs (miR3932b levels showed the greatest degree of reduction at -19.6-fold). Exposure of wild-type *Arabidopsis* whole seedlings to drought stress enhanced the abundance of 111 miRNAs and reduced the levels of 2 miRNAs, with the accumulation of miRNAs, miR851 (31.1-fold) and miR397b (-7.8-fold), determined to be influenced to the greatest degree by drought stress treatment. Post salt stress exposure, 86 miRNAs were determined to have a greater than 2-fold elevated abundance (miR778 was upregulated to the greatest degree at 34.0-fold), and further, 22 miRNAs were determined to be reduced in their abundance (miR169g showed the greatest degree of reduction at -8.7-fold). It was also of interest to document reciprocal abundance trends for an additional 5, 5 and 1 miRNA sRNAs upon comparison of each of the applied stresses, namely the heat/drought, heat/salt and drought/salt stress comparisons, respectively. In addition, a further 4 miRNA sRNAs, including miR169f, miR169h, miR397b and miR857, were determined to have an opposing change in abundance upon exposure to each of the three abiotic stresses assessed (Figure 2B). These findings indicate that the promoter regions of the encoding loci of these miRNAs harbor multiple *cis*-elements that direct the changes in *MIR* gene expression which would be required to result in the observed changes in the abundance of this miRNA cohort post exposure to different abiotic stresses [18,19,20].

A modified RT-qPCR approach [36] was next employed to experimentally validate the sequencing determined abundance of five miRNAs for each stress treatment. For heat stressed *Arabidopsis* whole seedlings, sequencing determined that the abundance of miRNAs, miR169, miR395 and miR396, was altered -7.4, 37.8 and 2.9-fold, respectively, compared to their abundance in non-stressed plants. The altered abundance trend of all three miRNAs was confirmed by RT-qPCR with quantified fold changes of -3.2-, 2.2- and 2.9-fold for the miR169, miR395 and miR396 sRNAs, respectively (Figure 3A–C). Although the abundance changes determined by RT-qPCR were not as dramatic as those determined via sequencing for miRNAs miR169 and miR395 (especially for miR395), the obtainment of a matching abundance trend for each quantified miRNA post heat stress exposure was highly encouraging. Therefore, RT-qPCR was again applied to confirm the sequencing identified abundance fold changes of -2.7-, 4.0- and 2.7-fold for miRNAs, miR857, miR156 and miR399, respectively (Figure 3D–F). Fold changes of -4.3-, 3.0- and 3.2-fold were determined for the miR857, miR156 and miR399 sRNAs respectively, post the 7-day drought stress treatment of *Arabidopsis* whole seedlings by RT-qPCR. RT-qPCR also confirmed the sequencing identified miRNA abundance trends for salt stressed *Arabidopsis* whole seedlings. Fold changes of -2.5-, 2.9- and 3.9-fold were determined by RT-qPCR for miRNAs, miR169, miR399 and miR778 respectively (Figure 3G–I), compared to the abundance fold changes of -4.8-, 4.0- and 34.0-fold determined via sequencing for these three miRNAs in response to salt stress treatment. In addition, miR839 and miR855 abundance in heat, drought and salt stressed plants was also quantified via RT-qPCR due to sequencing indicating that the level of both of these miRNAs did not vary significantly post application of each stress (Figure 3J,K). RT-qPCR confirmed that the levels of these two miRNAs varied less than 0.5-fold post stress exposure, a finding that suggests that neither miRNA is abiotic stress responsive. Taken together, the data presented in Figure 3 demonstrated that the high throughput sequencing employed here (Figure 2) was a reliable tool for profiling miRNA abundance changes in abiotic stressed *Arabidopsis* whole seedlings, and that once identified, RT-qPCR quantification provides a more biologically accurate reflection of the changes in sRNA abundance post exposure to each stress treatment.

### 2.3. Assessment of microRNA Target Gene Expression in Heat, Drought and Salt Stressed Arabidopsis Whole Seedlings

It has been extensively documented in *Arabidopsis* that miRNA sRNAs direct expression regulation of their targeted gene(s) via either a mRNA cleavage or translational repression mode of miRNA-directed RNA silencing [9,21,22,23]. Therefore, to identify the mode of target gene expression regulation directed by the miRNAs experimentally validated here to be responsive to heat, drought or salt stress, RT-qPCR was next employed to reveal the changes in miRNA target gene transcript abundance post exposure of *Arabidopsis* whole seedlings to these three abiotic stresses. For miRNAs, miR169, miR395 and miR396, the three miRNAs determined to be responsive to heat stress treatment via their RT-qPCR-determined, -3.2-, 2.2- and 2.9-fold change in abundance, the transcript level of a single target gene for each miRNA was demonstrated to have a reciprocally altered trend in abundance to that of their targeting miRNA (Figure 4A–C). Specifically, the miR169 target, *NUCLEAR FACTOR Y*, *SUBUNIT A5* (*NFYA5*; *AT1G54160*) was determined to have a 32.9-fold elevation in expression in heat stressed wild-type *Arabidopsis* compared to its levels in non-stressed whole seedlings (Figure 4A). In addition, *ATP SULFURYLASE1* (*ATPS1*; *AT3G22890*) and *GROWTH REGULATING FACTOR7* (*GRF7*; *AT5G53660*), the target genes of the heat-induced miRNAs, miR395 and miR396, respectively, were determined to have 2.4- (Figure 4B) and 2.2-fold (Figure 4C) reduced expression. RT-qPCR identified similar trends in altered expression for *LACCASE7* (*LAC7*; *AT3G09220*), *SQUAMOSA PROMOTER BINDING PROTEIN-LIKE9* (*SPL9*; *AT2G42200*) and *PHOSPHATE2* (*PHO2*; *AT2G33770*), the target genes of drought responsive miRNAs, miR857, miR156 and miR399, respectively. That is, *LAC7* expression was elevated 2.5-fold (Figure 4D) in response to the 2.7-fold reduction in miR857 levels (Figure 3D), and *SPL9* (Figure 4E) and *PHO2* (Figure 4F) expression was repressed by 2.8- and 4.5-fold respectively, in accordance with the documented 4.0- and 2.7-fold elevated abundance of the targeting miRNAs, miR156 (Figure 3E) and miR399 (Figure 3F), post drought stress treatment of *Arabidopsis* whole seedlings. 

Reciprocal expression trends were again observed post RT-qPCR assessment of target gene expression in salt stressed samples. Namely, *NFYA5* transcript abundance was significantly elevated 19.7-fold (Figure 4G) in response to the RT-qPCR documented 2.5-fold reduction in miR169 levels (Figure 3G). Further, the transcript abundance of *PHO2* and *SU(VAR)3-9 HOMOLOG6* (*SUVH6*; *AT2G22740*) was reduced by -2.8- (Figure 4H) and -6.6-fold (Figure 4I), respectively. Reduced *PHO2* and *SUVH6* expression in salt stressed *Arabidopsis* whole seedlings was not a surprising observation considering that the abundance of their targeting miRNAs, miR399 and miR778, was determined to be elevated by 2.9- and 3.9-fold (Figure 3H,I), respectively. Taken together, the target gene expression data presented in Figure 4 indeed suggested that altered miRNA abundance in response to each assessed stress was in turn leading to changes in miRNA target gene transcript abundance. In addition, demonstration of reciprocal trends in abundance for the miRNAs determined to be abiotic stress responsive in Figure 3 compared to the miRNA target gene expression profiles presented in Figure 4, strongly suggested that each abiotic stress responsive miRNA was regulating the expression of its assessed target gene via a mRNA cleavage mode of miRNA-directed RNA silencing.

### 2.4. Profiling of Salt Responsive microRNA Expression Modules in Arabidopsis Root and Shoot Tissues

To determine whether the documented alterations to abiotic stress responsive miRNA expression modules identified in *Arabidopsis* whole seedlings was an accurate indication of the changes occurring in specific and developmentally distinct *Arabidopsis* tissues, the three miRNA expression modules determined to be salt responsive in *Arabidopsis* whole seedlings (the miR169/*NFYA5*, miR399/*PHO2* and miR778/*SUVH6* expression modules), were profiled in *Arabidopsis* root and shoot tissue by RT-qPCR post exposure to salt stress. Prior to performing this molecular analysis however, the root architecture of wild-type *Arabidopsis* plants cultivated on vertically orientated control and salt stress growth media was assessed. It has been demonstrated previously that the major phenotypic response of the *Arabidopsis* root system to exposure to salt stress is reduced expansion of the primary root [37]. Figure 5A clearly shows that compared to non-stressed wild-type *Arabidopsis* seedlings, the primary phenotypic response of the root system of Col-0 plants exposed to the 7-day salt stress regime was inhibition of primary root elongation, with primary root length reduced by ~60% in salt stressed plants compared to the primary root length of non-stressed control plants (Figure 5A,B). 

Inhibition of primary root elongation, coupled with the vertically cultivated salt stressed plants again displaying reductions to the overall size of aerial tissue (i.e., rosette size; Figure 5A), as demonstrated in Figure 1A for *Arabidopsis* plants cultivated on horizontally orientated growth media, led us to next assess *P5SC1* expression in the vertically cultivated salt stressed *Arabidopsis* root and shoot tissue (Figure 5C). Compared to its levels in non-stressed roots and shoots, RT-qPCR revealed *P5CS1* expression to be significantly induced with transcript abundance elevated by 9.1- and 44.0-fold respectively, in salt stressed *Arabidopsis* roots and shoots (Figure 5C). This finding strongly suggested that both tissues types were indeed ‘*stressed*’ by the 7 days of vertical cultivation on plant growth media supplemented with 150 mM sodium chloride. RT-qPCR was therefore next employed to profile the miR169/*NFYA5*, miR399/*PHO2* and miR778/*SUVH6* expression modules in salt stressed root and shoot samples and revealed an opposing trend in abundance for each profiled expression module across both assessed tissues. For example, miR169 abundance was determined to be reduced by 1.6- and 2.2-fold in salt stressed roots and shoots respectively (Figure 5D), while *NFYA5* expression was elevated 4.6- and 16.5-fold in these two tissues (Figure 5E). Similar altered trends in abundance for the miR399 sRNA and its targeted transcript, the *PHO2* mRNA, were also revealed by RT-qPCR, namely; miR399 abundance was elevated by 3.7- and 3.0-fold in salt stressed roots and shoots (Figure 5F), and *PHO2* target gene expression was repressed accordingly in the corresponding tissues by 3.6- and 2.3-fold (Figure 5G), respectively. In addition, RT-qPCR revealed that the abundance of the *SUVH6*-targeting miRNA, miR778, was elevated in both salt stressed root and shoot tissue (Figure 5H). The 3.9- and 2.1-fold elevated abundance of the miR778 sRNA in *Arabidopsis* roots and shoots following the salt stress treatment was determined to result in repressed target gene expression, with the abundance of the *SUVH6* transcript reduced by ~2.0-fold in both assessed tissues (Figure 5I). Taken together, the data presented in Figure 5 confirmed that for the three miRNAs determined to be responsive to salt stress, via their profiling in *Arabidopsis* whole seedlings using a high throughput sequencing approach, provided an accurate reflection of the altered abundance of these three miRNAs in developmentally distinct tissues.

## 3. Discussion

In an attempt to provide sustainable food security into the future, it is essential that the complex, fundamental molecular networks that underpin the ability of a plant to maintain yield, particularly during extended periods of abiotic stress, are elucidated. This would provide the foundation for plant biology researchers to use a molecular approach to develop new plant lines that are readily able to adapt to, or mitigate the negative impacts that result from exposure to abiotic stress. With the miRNA class of sRNA demonstrated to be a key regulator of all aspects of plant development, as well as playing a central role in the ability of a plant to mount a defensive response against invading pathogens, or to mediate an adaptive response to abiotic stress, there currently remains a significant lack of resource datasets available for *Arabidopsis* to allow researchers to identify candidate miRNA expression modules for molecular modification as part of the future development of new plant lines that display adaptive phenotypes to abiotic stress. Towards this goal, here the genetic model plant species, *Arabidopsis thaliana*, was used to profile the miRNA landscape that potentially underpins, in part, the physiological and phenotypic responses of *Arabidopsis* to exposure to the abiotic stresses, heat, drought and salt stress. Most notably, sRNA sequencing revealed that the abundance of 121, 123 and 118 mature miRNA sRNAs was significantly (>2.0-fold) up- or down-regulated in response to heat, drought and salt stress treatment of *Arabidopsis* whole seedlings, respectively. The subsequent experimental validation of the miRNA abundance changes identified via high throughput sequencing by RT-qPCR, in combination with the additional use of RT-qPCR to document reciprocal trends in transcript abundance for each assessed miRNA target gene, was essential to confidently identify the miRNA expression modules responsive to each assessed stress.

In response to heat stress, a significant reduction to miR169 abundance (-3.2-fold) was observed in *Arabidopsis* whole seedlings (Figure 3A). Reduced miR169 levels have been reported previously for *Arabidopsis* post exposure to either drought stress or nitrogen starvation [38,39]. Further, [39] went on to demonstrate that reduced miR169 abundance in drought stressed *Arabidopsis* resulted in deregulated *NFYA5* expression, a target gene expression trend also observed here for heat stressed *Arabidopsis* (Figure 4A). The authors also revealed that overexpression of *NFYA5* in *Arabidopsis* resulted in these molecularly modified plant lines displaying reduced leaf water loss, due to reduced stomatal aperture, and drought stress tolerance [39]. Documentation of similar alterations to the miR169/*NFYA5* expression module in this study post heat stress treatment of *Arabidopsis* whole seedlings (Figure 3A, Figure 4A), to those previously reported for drought stressed *Arabidopsis* [39], suggests that these molecular changes are potentially in part driving the physical adaptation of *Arabidopsis* to both stresses, namely, alteration of stomatal aperture to promote water retention during exposure to such stress. It is also important to note here that the expression of several of the *MIR169* gene loci from which the miR169 precursor transcripts are transcribed were demonstrated to be induced in *Arabidopsis* by heat stress [40]. Induction of *MIR169* gene expression would be expected to result in elevated mature miR169 sRNA accumulation, and not reduced miR169 abundance, as observed here (Figure 3A). Curiously however, [40] did not report on whether *MIR169* gene expression induction actually resulted in elevated miR169 abundance in heat stressed *Arabidopsis* plants. The noted reduction to miR169 abundance reported here for heat stressed *Arabidopsis* plants, suggests that in *Arabidopsis*, heat stress represses *MIR169* gene expression, rather than promote transcription from these loci as reported by [40]. The opposing effect of heat stress exposure on miR169 accumulation in *Arabidopsis* reported here in Figure 3A, to that previously reported [40], could potentially be the result of differences in the application of the stress. Specifically, in [40], two-week-old *Arabidopsis* plants were transferred to moistened filter paper and exposed to the 40 °C heat stress treatment for a duration of either 3 or 6 h, whereas here, 8-day-old *Arabidopsis* seedlings were exposed to a prolonged 7-day heat stress treatment of elevated day/night (16/8 h) temperatures of 32 °C/28 °C. Nonetheless, the detection of elevated target gene (*NFYA5*) expression (Figure 4A), in accordance with the documented reduction in the abundance of the miR169 sRNA (Figure 3A), indicates that the alterations to the miR169/*NFYA5* expression module observed here in heat stressed *Arabidopsis*, are biologically relevant. 

Elevated abundance has previously been reported for miRNAs, miR395 and miR396, post exposure of *Arabidopsis* to heat stress [41,42]. Similar abundance changes for the miR395 (Figure 3B) and miR396 (Figure 3C) sRNAs were observed here for heat stressed wild-type *Arabidopsis* whole seedlings. In addition, elevated miRNA abundance was further demonstrated to direct enhanced miRNA-directed target gene expression repression, with both the miR395 and miR396 target genes, *ATPS1* (Figure 4B) and *GRF7* (Figure 4C) respectively, determined to have reduced transcript abundance in heat stressed *Arabidopsis*. Taken together, comparison of the findings reported here, to those reported previously for miR395 and miR396 [41,42], strongly suggest that these two miRNAs are indeed heat stress responsive miRNAs, and further, that enhanced miR395- and miR396-directed expression repression of *ATPS1* and *GRF7*, respectively, potentially forms part of the adaptive response of *Arabidopsis* to elevated temperature. 

Here, mannitol was used as an osmoticum to stimulate osmotic stress in *Arabidopsis* whole seedlings in an attempt to replicate drought stress conditions in a tightly controlled growth environment (i.e., sealed plant tissue culture plates). Post stress treatment, miR857 abundance was revealed to be reduced in *Arabidopsis* whole seedlings via both high throughput sequencing and RT-qPCR (Figure 2A, Figure 3D). The miR857 sRNA has previously been demonstrated to post-transcriptionally regulate the expression of *LAC7*, a laccase enzyme involved in mediating lignin deposition in the secondary xylem [43]. In addition to revealing reduced miR857 abundance, RT-qPCR showed that *LAC7* expression was elevated in drought stressed *Arabidopsis* whole seedlings (Figure 4D). Considering its documented role in secondary xylem development, the observed alterations to the miR857/*LAC7* expression module may potentially mediate an adaptive response to osmotic stress in *Arabidopsis*, potentially directing a change to tissue architecture in response to drought stress. Unlike miR857, miR156 and miR399 abundance was elevated by the mannitol-induced drought stress treatment (Figure 3E,F). In accordance, RT-qPCR showed that the transcript abundance of *SPL9* (Figure 4E) and *PHO2* (Figure 4F), the target genes of miR156 and miR399, respectively, was reduced in response to the elevated abundance of their targeting miRNA sRNAs. Interestingly, the miR156/*SPL9* expression module, together with the downstream gene, *DIHYDROFLAVONOL-4-REDUCTASE* (*DFR*; *AT5G42800*) have been demonstrated previously to play a role in anthocyanin metabolism [44], and Figure 1D shows that anthocyanin accumulation remained at its non-stressed levels in *Arabidopsis* plants cultivated on standard growth media supplemented with 200 mM mannitol, in spite of these plants displaying reductions to their fresh weight and rosette diameter, in addition to elevated expression of the stress induced gene, *P5CS1* (Figure 1E). Furthermore, in *Arabidopsis*, both of these miRNAs (miR156 and miR399) have been previously demonstrated to be responsive to mannitol-induced drought stress [41,44], findings that when taken together with those presented here in Figure 3 and Figure 4, strongly suggest that the miR156/*SPL9* and miR399/*PHO2* expression modules are indeed responsive to mannitol-induced drought stress. 

High throughput sRNA sequencing and RT-qPCR revealed miR169 abundance to be reduced by -4.8 and -2.5-fold respectively, post exposure to salt stress. This abundance change opposes that reported previously for the miR169 sRNA in rice and cotton [45,46], where miR169 accumulation was demonstrated to be induced by salt stress. However, the observed differences in miR169 abundance post salt stress exposure in rice [45], cotton [46] and *Arabidopsis* (Figure 3G), is most likely the result of unique *cis*-element landscapes of the promoter regions of *MIR169* loci across these three species [19]. In *Arabidopsis*, miR169 abundance has been previously demonstrated to be reduced by drought stress [40], conditions of limited phosphate [47], and nitrogen starvation [38]. The findings of these reports [38,40,47], together with those presented here, namely deregulated *NFYA5* target gene expression in *Arabidopsis* whole seedlings (Figure 4G), roots and shoots (Figure 5E), due to loss of miR169-directed *NFYA5* expression repression in these tissues, indicates that the miR169/*NFYA5* expression module potentially plays a central role in mediating the response of *Arabidopsis* to a range of abiotic stresses, potentially even forming a ‘*crosstalk junction*’ to link the highly complicated molecular networks that are required to directed the physiological and phenotypic responses of *Arabidopsis* to abiotic stress.

Salt stress treatment was shown to enhance miR399 sRNA abundance in *Arabidopsis* whole seedlings (Figure 3H) as well as in root and shoot tissue (Figure 5F), a previously reported finding [41]. Furthermore, and using a molecular approach in *Arabidopsis*, [48] revealed *MIR399F* gene expression to be induced by salt stress and that *Arabidopsis* plants modified to constitutively overexpress the *MIR399F* gene were more tolerant to salt stress than unmodified wild-type plants. Given *PHO2* targeting by miR399, and the previously documented role for phosphate in modulating root system architecture alterations under salt stress conditions in *Arabidopsis,* the elevated abundance of miR399 shown here, in conjunction with the demonstrated reductions to the level of the *PHO2* target transcript (Figure 4H, Figure 5G), are consistent with the proposed role of the miR399/*PHO2* expression module in the complex phosphate-salt regulatory network in *Arabidopsis* tissues [49,50]. Like miR399, miR778 has previously been classed as a phosphate responsive miRNA in *Arabidopsis* [47,51,52]. Here we demonstrate that miR778 abundance is also elevated in response to salt stress in *Arabidopsis* whole seedlings (Figure 3I), roots and shoots (Figure 5H). Accordingly, via a RT-qPCR approach, we further revealed that elevated miR778 abundance resulted in enhanced expression repression of the miR778 target gene, *SUVH6*, in salt stressed *Arabidopsis* tissues (Figure 4I, Figure 5I). Interestingly, the miR778 target, SUVH6, is involved in directing methylation of the lysine 9 residue of histone H3 (H3K9 methylation), and further, *SUVH6* expression repression via the constitutive overexpression of the *MIR778* precursor transcript resulted in the modified *Arabidopsis* plants displaying moderately enhanced primary and lateral root growth, and elevated levels of free phosphate and anthocyanin accumulation in the aerial tissues of these plants when cultivated in a phosphate deficient growth environment [52]. These findings, together with the alterations to both the miR399/*PHO2* and miR778/*SUVH6* expression module reported here for salt stressed *Arabidopsis* (Figure 3 and Figure 5), add further weight to the importance of phosphate-mediated responses in *Arabidopsis* tissues as part of the adaptive response of *Arabidopsis* to salt stress.

Altered miRNA abundance, and miRNA target gene expression, have been identified as key molecular responses to an array of abiotic stresses across an evolutionary diverse range of plant species [6,14,24,41,48,49]. Here we have specifically assessed alterations to the miRNA landscapes of heat, drought and salt stressed wild-type *Arabidopsis* whole seedlings and identified large miRNA cohorts responsive to each stress. Alteration to a select number of miRNA/target gene expression modules for the heat, drought and salt stress treatments were experimentally validated via an RT-qPCR approach. Considering that many abiotic responsive miRNAs have been demonstrated to play a conserved functional role across phylogenetically diverse plant species [14,24,25], it is envisaged that the dataset generated in this study forms a valuable resource for the wider plant biology research community; a resource that can be used as the starting point to identify the specific miRNA expression modules to be molecularly manipulated in plant species amenable to genetic modification as part of the future development of plant lines with an altered miRNA and/or miRNA target gene abundance that display a tolerance phenotype to either heat, drought or salt stress. Alternatively, for plant species that are not readily amenable to genetic modification, this dataset can additionally be used to identify the specific miRNA expression modules to be targeted for rapid high throughput screening (via RT-qPCR) across diverse germplasm of a specific species to select those genotypes that harbor natural alterations to the molecular profile of the miRNA expression module of interest.

## 4. Materials and Methods 

### 4.1. Plant Material

The seeds of wild-type *Arabidopsis thaliana* (*Arabidopsis*), ecotype Columbia-0 (Col-0), were surface sterilized using chlorine gas and post sterilization, seeds were plated out onto standard *Arabidopsis* plant growth media (half strength Murashige and Skoog (MS) salts) and stratified in the dark at 4°C for 48 h. Post stratification, sealed plates containing the surface sterilized seeds were transferred to a temperature-controlled growth cabinet (A1000 Growth Chamber, Conviron^®^ Australia) and cultivated for 8 days under a standard growth regime of 16 h light/8 h dark and a 22 °C/18 °C day/night temperature. Following this 8-day cultivation period, equal numbers of Col-0 seedlings were transferred under sterile conditions to either fresh standard *Arabidopsis* plant growth media (control treatment), or to plant growth media that had been supplemented with 200 millimolar (mM) mannitol (drought stress treatment) or 150 mM of sodium chloride (salt stress treatment). Post seedling transfer, the non-stressed control and the drought and salt stress treatment plates were returned to the growth cabinet and cultivated for an additional 7-day period under standard growth conditions. For the heat stress treatment, 8-day-old seedlings were also transferred under sterile conditions to standard growth media, however the 16/8 h day/night temperature was elevated to 32 °C/28 °C for the duration of the 7-day stress treatment period. At the end of the 7-day treatment period, all of the phenotypic and molecular assessments reported here were conducted on 15 day old *Arabidopsis* whole seedlings. For the tissue specific analyses reported in Figure 5, plants were treated exactly as outlined above, except for the 7-day treatment period, when 8 day old seedlings were transferred and cultivated on control and salt stress media plates that were orientated for vertical growth.

### 4.2. Phenotypic and Physiological Assessments

All phenotypic assessments reported here were conducted on 15 day old *Arabidopsis* seedlings. The performance of wild-type *Arabidopsis* plants exposed to each assessed stress is therefore presented relative to non-stressed control plants. More specifically, each phenotypic measurement collected for *Arabidopsis* seedlings exposed to each assessed abiotic stress regime is presented as a percentage of the corresponding measurement determined for non-stressed control seedlings cultivated under standard growth conditions for the duration of the 7-day stress treatment period. Rosette diameter and primary root length analysis was determined via assessment of photographic images using the ImageJ software. A standard 99:1 (v/v) methanol:HCl extraction protocol was used to extract anthocyanin from control and stress treated Col-0 plants. Post extraction, anthocyanin content was determined using a spectrophotometer (Thermo Scientific, Australia) at an absorbance wavelength of 535 nanometers (A_535_) and using the 99:1 (v/v) methanol:HCl solution as the blank.

### 4.3. Total RNA Extraction and High Throughput Sequencing of the small RNA Fraction

Total RNA was isolated from four biological replicates (each biological replicate contained 6 individual plants) of 15 day old Col-0 whole seedlings cultivated under normal growth conditions for the duration of the experimental period, or post 7-days of heat, drought or salt stress treatment, using TRIzol™ Reagent (Invitrogen™) according to the manufacturer’s instructions. The quality and quantity of the isolated total RNA was assessed using a Nanodrop spectrophotometer (NanoDrop^®^ ND-1000, Thermo Scientific, Australia) and via standard electrophoresis on a 1.2% (w/v) ethidium bromide-stained agarose gel to allow for RNA visualization. Next, 5.0 micrograms (µg) of each of the four biological replicates for each treatment, were pooled together and diluted in RNase-free water to obtain a final preparation of 25 microliters (μL) of total RNA at a concentration of 800 nanograms (ng) per µL. Samples were shipped to the Australian Genome Research Facility (AGRF; Melbourne node, Australia) with the AGRF performing all subsequent preparatory steps prior to sequencing the small RNA fraction of each sample on an Illumina HiSeq 2500 platform. 

### 4.4. Bioinformatic Assessment of the microRNA Landscape of Arabidopsis Whole Seedlings

Using the Qiagen CLC Genomics Workbench (11) software, next-generation sequencing adapter sequences were removed prior to performing sequence quality trimming to remove any sRNA reads that were either shorter than 15 nucleotides (nts), or longer than 35 nts in length. Additionally, parameters within the CLC Genomic Workbench were applied to remove any ambiguous nucleotides at either the 5’ or 3’ terminus of each sequencing read (i.e., the removal of any ‘N’ nucleotides on sequence ends), or to ‘trim’ low quality sequences using a modified ‘*Mott trimming*’ algorithm. The remaining sequences that aligned perfectly (i.e., zero mismatches) to known *Arabidopsis* miRNAs listed in miRBase 22 were then annotated. The values determined for the; (1) raw read count of each detected miRNA sRNA across the four treatments (control, heat, drought and salt); (2) Log2 fold change in abundance for each miRNA sRNA per stress treatment, compared to the non-stressed control values; (3) total number of high quality raw reads per library; (4) total number of miRNA sRNA raw reads per library, and; (5) percentage of the total library size that the miRNA class of sRNA represents, is presented in Appendix A.

### 4.5. Quantitative Reverse Transcriptase Polymerase Chain Reaction Analyses

Quantitative reverse transcriptase polymerase chain reaction (RT-qPCR) assessment of miRNA sRNA and miRNA target gene transcript abundance was conducted on 4 biological replicates: the same four biological replicates that were pooled together to perform the high throughput sequencing analysis of the sRNA fraction of each sample. The synthesis of miRNA-specific complementary DNA (cDNA) was conducted using 200 ng of DNase I treated (New England BioLabs, Australia) total RNA as template and 1.0 unit (U) of ProtoScript^®^ II Reverse Transcriptase (New England BioLabs, Australia) according to manufacturer’s instructions. The cycling conditions for miRNA-specific cDNA synthesis were: 1 cycle at 16°C for 30 min; 60 cycles of 30°C for 30 s, 42°C for 30 s 50°C for 2 s, and; 1 cycle of 85°C for 5 min. To generate a global high molecular weight cDNA library for the quantification of miRNA target gene expression, 5.0 µg of total RNA was treated with 5.0 U of DNase I (New England BioLabs, Australia) according to manufacturer’s instructions. Post DNase I treatment, the total RNA was purified using an RNeasy Mini Kit according to the manufacturer’s instructions (Qiagen, Australia), and then 1.0 μg of this preparation was used as template for cDNA synthesis with 1.0 U of ProtoScript^®^ II Reverse Transcriptase according to the manufacturer’s instructions (New England Biolabs, Australia) along with 2.5 μM of oligo dT_(18)_. All single stranded cDNA preparations were next diluted to 50 ng/μL in RNase-free water prior to RT-qPCR quantification of miRNA sRNA abundance or miRNA target gene expression. The GoTaq^®^ qPCR Master Mix (Promega, Australia) was used as the fluorescent reagent for all performed RT-qPCRs, and all RT-qPCRs had the same cycling conditions of: 1 cycle of 95°C for 10 min, followed by 45 cycles of 95°C for 10 s and 60°C for 15 s. The abundance of each assessed miRNA sRNA and the expression of each examined miRNA target gene was determined using the 2^−ΔΔCT^ method with the small nucleolar RNA, *snoR101*, and *UBIQUITIN10* (*UBI10*; *AT4G05320*) used as the respective internal controls to normalize the relative abundance of each assessed transcript. All DNA oligonucleotides used for either miRNA-specific cDNA synthesis or the quantification of miRNA target gene expression are provided in Appendix A. For the synthesis of miRNA-specific cDNA of a miRNA sRNA that belongs to a multimember family, and where multiple family members were detected via the high throughput sequencing approach, a miRNA family consensus sequence was determined and the primer designed to hybridize will all detected family members.

## Figures and Tables

**Figure 1 plants-08-00058-f001:**
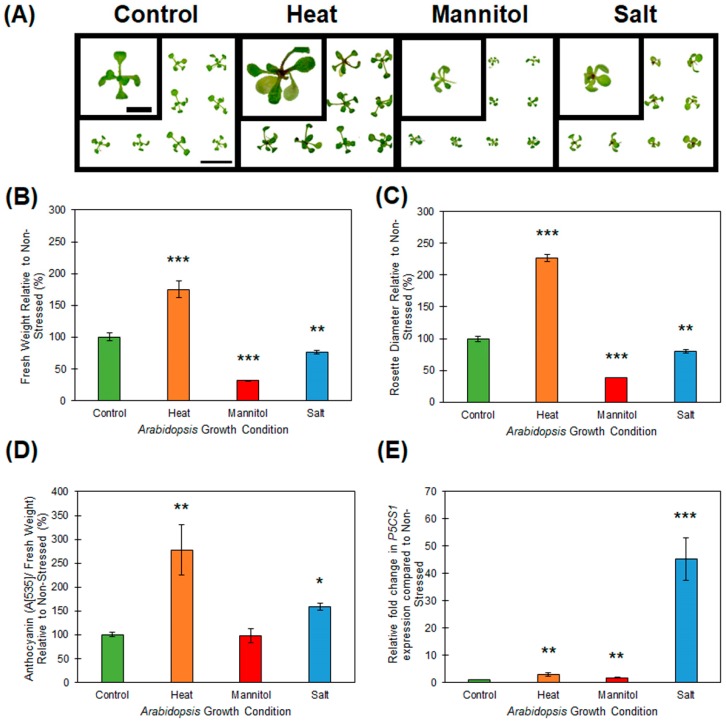
Phenotypic and physiological consequence of heat, drought and salt stress treatment of 15-day-old wild-type *Arabidopsis* whole seedlings. (**A**) Phenotypes displayed by 15-day old wild-type *Arabidopsis* whole seedlings post a 7-day treatment regime with heat, drought or salt stress, compared to non-stressed seedlings of the same age (left panel). Scale bar = 1.0 centimeter (cm) on larger sized panels and 0.5 cm on the superimposed images of a single representative seedling. (**B**) Whole seedling fresh weight of heat, drought (mannitol) and salt stressed *Arabidopsis* compared to their non-stressed counterparts of the same age. (**C**) Rosette diameter of 15-day-old *Arabidopsis* whole seedlings post 7-day exposure to heat, drought (mannitol) and salt stress compared to the non-stressed control. (**D**) Anthocyanin accumulation in heat, drought (mannitol) and salt stressed *Arabidopsis* whole seedlings compared to non-stressed whole seedlings of the same age (15 days). (**E**) RT-qPCR assessment of the expression of the stress induced gene, *Δ1-PYRROLINE-5-CARBOXYLATE SYNTHETASE1* (*P5CS1*; *AT2G39800*) expression in 15-day-old *Arabidopsis* whole seedlings post a 7-day heat, drought (mannitol) and salt stress treatment regime compared to the abundance of the *P5CS1* transcript in non-stressed *Arabidopsis* whole seedlings of the same age. (**B**–**E**) Error bars represent the standard deviation of four biological replicates and each biological replicate consisted of a pool of six individual plants. The presence of an asterisk above a column represents a statistically significantly difference between the stress treated sample and the non-stressed control sample (*p*-value: * < 0.05; ** < 0.005; *** < 0.001).

**Figure 2 plants-08-00058-f002:**
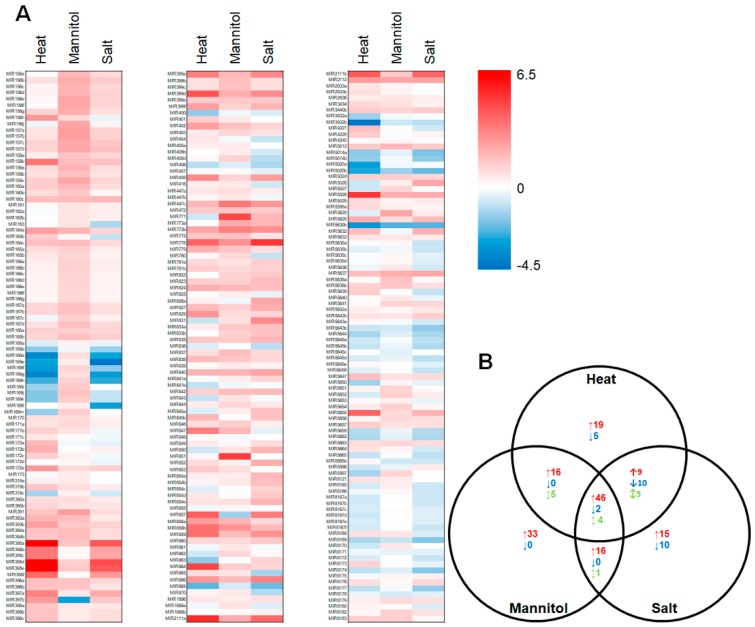
Profiling of the miRNA landscape of heat, drought and salt stressed 15-day-old wild-type *Arabidopsis* whole seedlings. (**A**) Red (up) and blue (down) shaded tiles represent a Log2 fold change in abundance of the *Arabidopsis* miRNA sRNAs detected via high throughput sequencing (see Appendix A for the normalized read numbers used to determine fold change values). (**B**) The number of miRNA sRNAs determined to have a greater than 2-fold change in abundance in heat, drought and salt stressed 15-day-old wild-type *Arabidopsis* whole seedlings compared to the abundance of each detected miRNA sRNA in non-stressed control plants of the same age. Red colored up arrows indicate the number of miRNAs with elevated abundance under each assessed stress, blue colored down arrows represent the number of miRNAs with reduced abundance post stress treatment and green colored up/down arrows state the number of miRNA sRNAs with a differing abundance trend between the individual stress treatments.

**Figure 3 plants-08-00058-f003:**
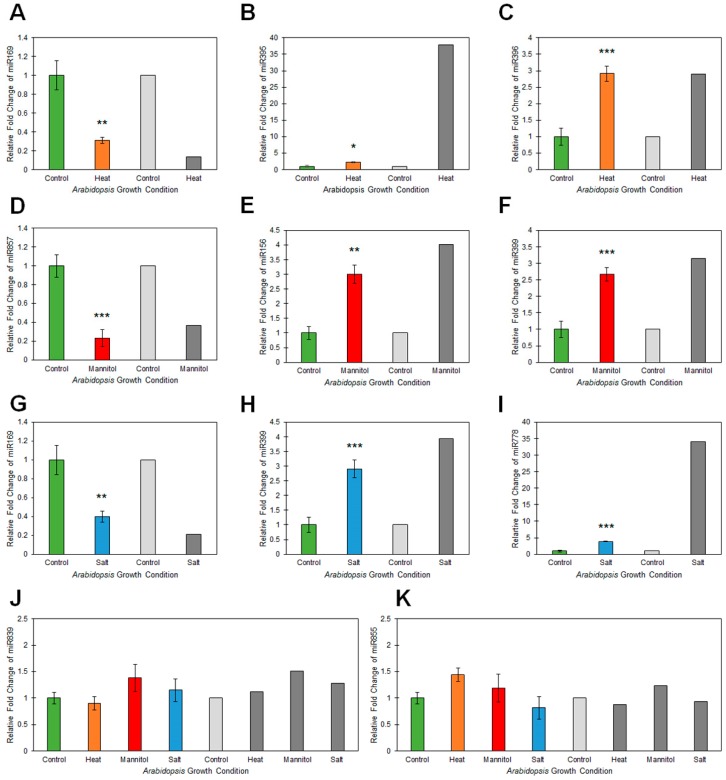
Quantification of miRNA abundance via RT-qPCR analysis of 15-day-old wild-type *Arabidopsis* whole seedlings post exposure to heat, drought and salt stress treatment. (**A**–**C**) RT-qPCR assessment of miR169 (**A**), miR395 (**B**) and miR396 (**C**) abundance in heat stressed *Arabidopsis* whole seedlings. (**D**–**F**) RT-qPCR assessment of miR857 (**D**), miR156 (**E**) and miR399 (**F**) abundance in drought (mannitol) stressed *Arabidopsis* whole seedlings. (**G**–**I**) RT-qPCR assessment of miR169 (**G**), miR399 (**H**) and miR778 (**I**) abundance in salt stressed *Arabidopsis* whole seedlings. (**J**,**K**) RT-qPCR assessment of miR839 (**J**) and miR855 (**K**) abundance across heat, drought (mannitol) and salt stressed *Arabidopsis* whole seedlings. (**A**–**K**) Colored columns (green = non-stressed control; orange = heat stress; red = drought stress, and; blue = salt stress) represent RT-qPCR determined abundance of each quantified miRNA sRNA and the light (control) and dark grey (stress) shaded columns present the fold changes in miRNA abundance as determined via high throughput sequencing. Error bars represent the standard deviation of four biological replicates and each biological replicate consisted of a pool of six individual plants. The presence of an asterisk above a column represents a statistically significantly difference between the stress treated sample and the non-stressed control sample (*p*-value: * < 0.05; ** < 0.005; *** < 0.001).

**Figure 4 plants-08-00058-f004:**
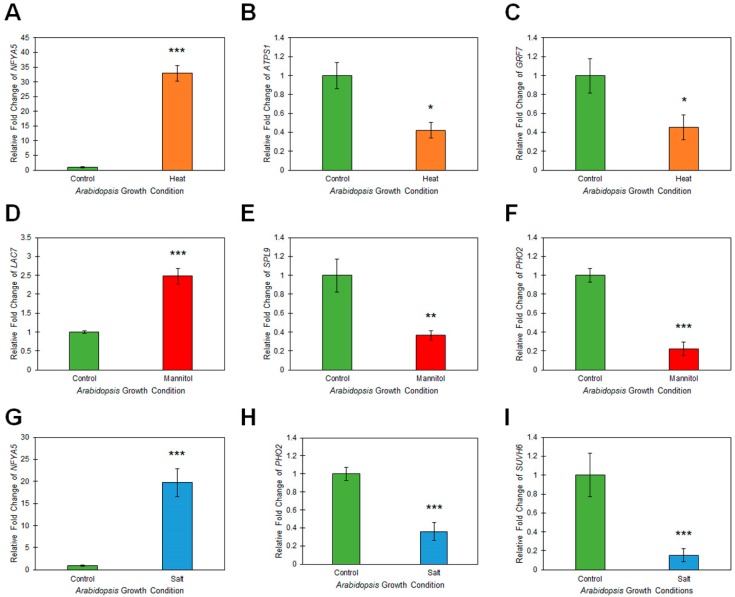
Determination of miRNA target gene expression via RT-qPCR analysis of 15-day-old wild-type *Arabidopsis* whole seedlings post exposure to heat, drought and salt stress treatment. (**A**–**C)** RT-qPCR assessment of *NFYA5* (**A**), *ATPS1* (**B**) and *GRF7* (**C**) miRNA target gene expression in heat stressed *Arabidopsis* whole seedlings. (**D**–**F**) RT-qPCR assessment of *LAC7* (**D**), *SPL9* (**E**) and *PHO2* (**F**) miRNA target gene expression in drought (mannitol) stressed *Arabidopsis* whole seedlings. (**G**–**I)** RT-qPCR assessment of *NFYA5* (**G**), *PHO2* (**H**) and *SUVH6* (**I**) miRNA target gene expression in salt stressed *Arabidopsis* whole seedlings. (**A**–**I**) Colored columns (green = non-stressed control; orange = heat stress; red = drought stress, and; blue = salt stress) represent RT-qPCR quantified expression of a single target gene for each miRNA assessed via RT-qPCR analysis in Figure 3. Error bars represent the standard deviation of four biological replicates and each biological replicate consisted of a pool of six individual plants. The presence of an asterisk above a column represents a statistically significantly difference between the stress treated sample and the non-stressed control sample (*p*-value: * < 0.05; ** < 0.005; *** < 0.001).

**Figure 5 plants-08-00058-f005:**
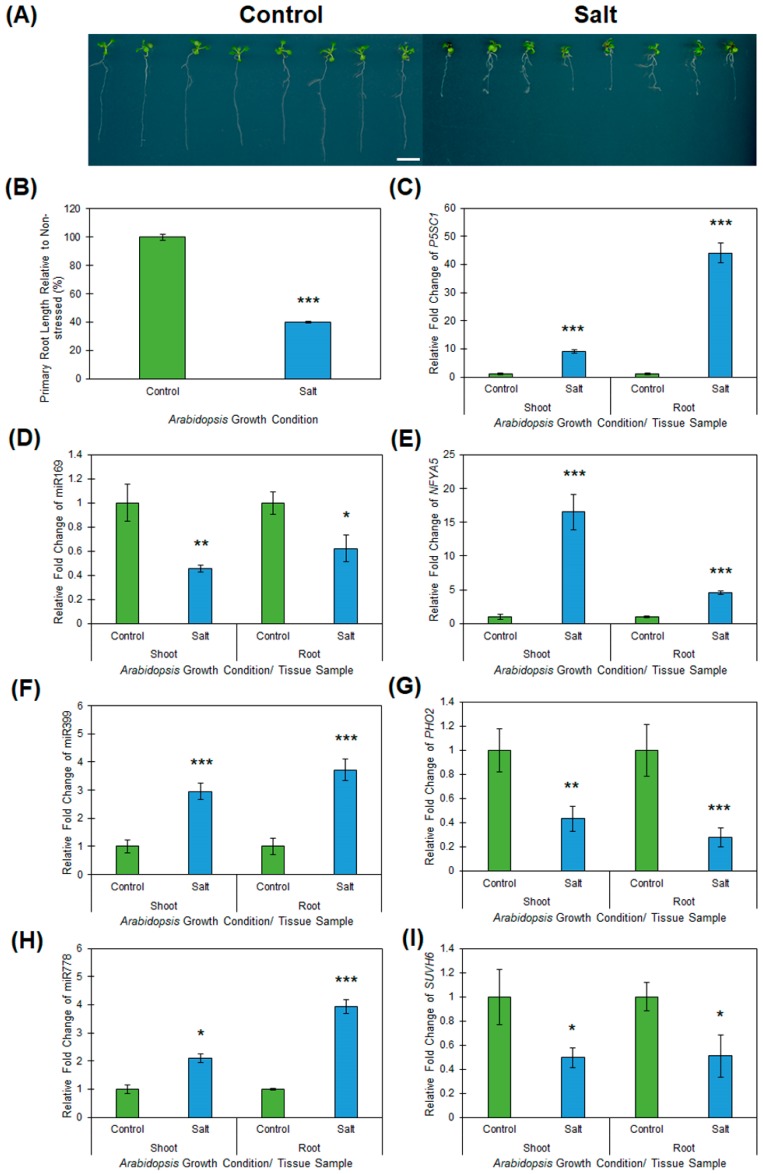
Phenotypic and molecular assessment of the root and shoot tissues of 15-day-old wild-type *Arabidopsis* plants post the 7-day salt stress treatment regime. (**A**) Root and shoot architecture of 15-day-old wild-type *Arabidopsis* seedlings post a 7-day salt stress treatment (right panel) during which the growth media plates were orientated for vertical growth. Scale bar = 1.0 cm. (**B**) Primary root length of 15-day-old *Arabidopsis* whole seedlings cultivated on vertically oriented media growth plates that contained either standard *Arabidopsis* growth media (non-stressed control) or growth media that had been supplemented with 150 mM sodium chloride (stress treatment). (**C**) RT-qPCR assessment of the expression of the stress induced gene, *P5CS1*, expression in 15-day-old *Arabidopsis* root and shoot material post 7-day salt stress treatment compared to the abundance of the *P5CS1* transcript in non-stress control plants of the same age. (**D**,**E**) RT-qPCR quantification of miR169 abundance (**D**) and *NFYA5* target gene expression (**E**) in salt stressed *Arabidopsis* root and shoot tissues. (**F**,**G**) RT-qPCR quantification of miR399 abundance (**F**) and *PHO2* target gene expression (**G**) in salt stressed *Arabidopsis* root and shoot tissues. (**H**,**I**) RT-qPCR quantification of miR778 abundance (**H**) and *SUVH6* target gene expression (**I**) in salt stressed *Arabidopsis* root and shoot tissues. (**B**–**I**) Colored columns represent the values obtained for non-stressed control plants (green colored columns) and the salt stressed samples (blue colored columns). Error bars represent the standard deviation of four biological replicates and each biological replicate consisted of a pool of six individual plants. The presence of an asterisk above a column represents a statistically significantly difference between the salt stress sample and the non-stressed controls (*p*-value: * < 0.05; ** < 0.005; *** < 0.001).

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
