# Peer review of "Profiling the Abiotic Stress Responsive microRNA Landscape of Arabidopsis thaliana"

_plants, 2019, doi:10.3390/plants8030058_

Round 1
Reviewer 1 Report
Pegler et al. used the high throughput sequencing method to profile the miRNA landscape of Arabidopsis under heat, drought, and salt stresses. They identified several miRNAs with a greater than 2-fold altered abundance after stress treatments based on NGS data. Then, some miRNAs were selected for experimental validation. However, the novelty and scientific soundness are low in this study. Because many validated genes were known to be affected under stresses previously. In addition, all miR/target gene pairs validated in this study are already known. The authors should more indicate their novel findings. Several points should also be addressed.
1. The size of Arabidopsis seedlings are increased after heat treatment in this study. However, I usually saw smaller Arabidopsis under heat stress in many previous studies. Did the author ever seen similar results in other papers? The related references could be applied.
2. The control should be added in the Figure 1.
3. Why the author select P5CS1 as stress index gene? As I know, P5CS1 is induced by abscisic acid and salt stress in a light-dependent manner. Does P5CS1 also involve in heat stress responsiveness? The related reference should be added.
4. How to select miRNAs for experimental validation? The authors should describe why and how to select those miRNA for RT-qPCR in the manuscript. Why didn’t they select some miRNAs which are specifically induced by heat, such as mir5655 and mir5028. Or, miR-851 is specifically induced by mannitol. Based on Figure 2A, mir156 is not specific for mannitol, but they select mir-156 to test under mannitol. Similar event is in the case of mir-399.
Author Response
Dear Editor,
On behalf of the authorship team, I would like to take this opportunity to thank the two Reviewers for taking the time to comprehensively review our manuscript submitted for consideration for inclusion in the upcoming Special Issue of Plants. After careful consideration of each of the points raised by each Reviewer, we provide our responses below. I hope you find that we have adequately addressed the Reviewers’ concerns in our revised manuscript. Please note; for those points of concern raised by either Reviewer that we have not addressed in the revised version of our manuscript, we provide detailed justification as to why below.
Regards,
Andrew Eamens.
Reviewer #1 comments for plants - 450616
Comments and Suggestions for Authors
Pegler et al. used the high throughput sequencing method to profile the miRNA landscape of Arabidopsis under heat, drought, and salt stresses. They identified several miRNAs with a greater than 2-fold altered abundance after stress treatments based on NGS data. Then, some miRNAs were selected for experimental validation. However, the novelty and scientific soundness are low in this study. Because many validated genes were known to be affected under stresses previously. In addition, all miR/target gene pairs validated in this study are already known. The authors should more indicate their novel findings. Several points should also be addressed.
1.1. The degree of novelty of the presented results is low.
1.2. The size of Arabidopsis seedlings are increased after heat treatment in this study. However, I usually saw smaller Arabidopsis under heat stress in many previous studies. Did the author ever seen similar results in other papers? The related references could be applied.
1.3. The control should be added in the Figure 1.
1.4. Why the author select P5CS1 as stress index gene? As I know, P5CS1 is induced by abscisic acid and salt stress in a light-dependent manner. Does P5CS1 also involve in heat stress responsiveness? The related reference should be added.
1.5. How to select miRNAs for experimental validation? The authors should describe why and how to select those miRNA for RT-qPCR in the manuscript. Why didn’t they select some miRNAs which are specifically induced by heat, such as mir5655 and mir5028. Or, miR-851 is specifically induced by mannitol. Based on Figure 2A, mir156 is not specific for mannitol, but they select mir-156 to test under mannitol. Similar event is in the case of mir-399.
Author responses to Reviewer #1 comments;
We thank Reviewer #1 for their constructive review of our manuscript and we have addressed the concerns of Reviewer #1 below in an attempt to improve our manuscript.
1.1. The primary aim of this study was to provide the Arabidopsis miRNA research community with a currently lacking experimental resource dataset, specifically: miRNA profiling datasets post exposure of wild-type Arabidopsis plants to heat, drought and salt stress. Considering the extensive utilisation of Arabidopsis as an experimental model for molecular studies for many years worldwide, it is highly surprising that such resource dataset is largely currently lacking. Specifically, searching of the PubMed catalogue (https://www.ncbi.nlm. nih.gov/pubmed/) using the combined search terms (search terms combined via the use of capital ‘AND’), ‘miRNA AND Arabidopsis AND salt stress’, ‘miRNA AND Arabidopsis AND drought stress’, and ‘miRNA AND Arabidopsis AND heat stress’ returns 39, 53 and 23 articles, respectively. Of the 115 identified articles, only a single study (PMID: 26089831) has previously utilised a small RNA sequencing approach to document miRNA accumulation profiles in Arabidopsis post abiotic stress exposure. Here we experimentally validated different miRNAs to those assessed in (PMID: 26089831) as well as to present expression trends for a target gene of each experimentally validated miRNA. In addition, the originally submitted manuscript already acknowledged this previously published study (PMID: 26089831) to compare and contrast the results presented here to those reported previously. We therefore find this comment by Reviewer #1 quite curious.
1.2. We thank Reviewer #1 for offering their own experience with cultivating Arabidopsis under conditions of elevated temperature. And yes, initially we were surprised with the observed ‘promotion’ of Arabidopsis growth during the heat stress regime that we applied in our study. Due to this unexpected result, we repeated this experiment numerous times, and each time, an identical ‘growth promotion’ phenotype was displayed by wild-type Arabidopsis plants cultivated under conditions of elevated temperature. It is important to note here, that we additionally repeated the drought and salt stress growth experiments numerous times to ensure accurate reporting of all phenotypic/physiological responses of Arabidopsis to each applied stress. Furthermore, we have searched the literature and identified two previously published studies (PMID: 9618562; PMID: 19249207) that also report the promotion of specific growth parameters for Arabidopsis plants cultivated under conditions of elevated temperature. Both of these studies, and a sentence outlining the similarity of the phenotypes reported in these two studies to that reported here for heat stressed Arabidopsis plants has been added to the text of the revised version of the manuscript (see lines 123-126 in revised manuscript).
1.3. The authorship team again thank Reviewer #1 for highlight this oversight in the originally submitted version of our manuscript. To address this oversight, we have generated a new version of Figure 1, with this updated version of Figure 1 including the non-stressed control measurements to which the other assessed parameters depicted in this Figure for heat, drought and salt stressed plants were compared (see lines 136-137 in the revised manuscript).
1.4. We again thank Reviewer #1 for also offering this helpful assessment of our originally submitted manuscript. P5CS1was selected for use as the marker of stress at the molecular level in this study due to the previous demonstrations that P5CS1 is responsive to heat stress (PMID: 21670222), dehydration (PMID: 19036030) and salt stress (PMID: 9351242). These three studies are cited in the revised version of our manuscript (see lines 133 and 158/159) and included in the updated reference list.
1.5. As stated in Response 1.1., the primary aim of this study was to provide a currently lacking experimental resource for the Arabidopsis miRNA research community (and the plant biology community in general), that is; provision of heat, drought and salt stress responsive miRNA datasets. We therefore selected well characterised miRNA/miRNA target gene expression modules to experimentally validate as abiotic stress responsive. More specifically, only (1) conserved (2) abundance (3) well characterised miRNAs with known target genes were selected for experimental validation post small RNA sequencing. Therefore, due to these criteria, we did not experimentally validate those specific miRNAs identified by Reviewer #1 for the following reasons;
miR851: based on PubMed search results, miR851 has only been reported in a single study in rice (PMID: 20729483) and BLASTn search analysis of the mature Ath-miR851sequence against the Arabidopsis TAIR10 transcriptome (https://www.arabidopsis.org /Blast/index.jsp) fails to identify any likely target transcripts. Furthermore, miR851 is a very lowly abundance miRNA at 2, 2, 61 and 0 reads in the non-stressed, heat, drought and salt stressed libraries; read numbers expected to be below the reliable detection sensitivity levels of our RT-qPCR approach in 3/4 libraries.
miR5028: based on PubMed search results, miR5028 has only been reported by a single study (PMID: 21357774), has not been demonstrated heat responsive previously, nor responsive to any other stress. Further, BLASTn search analysis of the mature miR5028 sequence against the Arabidopsis TAIR10 transcriptome (https://www.arabidopsis.org /Blast/index.jsp) fails to identify any likely target transcripts. In addition, sequencing demonstrated the miR5028 sRNA to be lowly abundant (non-stressed = 4, heat = 115, drought = 17 and salt = 17) in 3/4 of the generated sequencing libraries, a read abundance that was considered approaching or even below the level of detection sensitivity of our RT-qPCR approach.
miR5655: miR5655 is only reported in a single study (PMID: 21940835) on PubMed. Furthermore, no functional characterisation exists for this miRNA and BLASTn search the miR5655 sequence against the Arabidopsis TAIR10 transcriptome (https://www.arabidopsis.org /Blast/index.jsp) fails to identify any targeted transcripts. In addition, read numbers of 4, 55, 8 and 15 for the non-stressed, heat, drought and salt stressed libraries respectively, indicated that this miRNA would be approaching or below the detection sensitivities of our RT-qPCR approach.
Reviewer 2 Report
The study profiled the Arabidopsis microRNAome under drought, heat and salt stress. qPCR was used to analyse miRNA-target expression profile. The presentation of the results needs to be improved. Given that Arabidopsis microRNAs have been relatively well-documented in previous studies, the novelty and the highlights of this research need to be strengthened.
Line 83: Some good reference on utilising small RNAs for crop improvement. DOI: 10.1016/j.plantsci.2018.05.031; DOI: 10.1016/j.tplants.2016.07.006.
Line 87: Not really sure if this can be considered the first step, given that many studies have conducted sequencing analysis of Arabidopsis microRNAome under abiotic stress.
Lines 102-104: I don’t think there is enough evidence to support this statement. There is no molecular modification nor functional charaterisation?
Lines 153-155: The change of one gene doesn’t really suggest changes of the whole global RNA expression.
Lines 531 & 563: Was total RNA treated with DNase? Complete removal of genomic DNA is critical for sRNA sequencing and qPCR to ensure accuracy.
Line 550: How many mismatches were allowed for miRNA annotation?
Line 553: miRNAs with low RPM (normally less than 10) should be omitted from fold-change analysis. See examples. DOI: 10.1186/s12870-014-0196-4; DOI: 10.1371/journal.pone.0142799; DOI: 10.1371/journal.pone.0139658
The fold-change of some miRNAs (e.g. miR168k) in table S1 should not be included.
Line 152 onwards: Figure 2 and the Results section should be revised after removing the miRNAs with low RPM.
Lines 162-170: None of the fold-change examples matches the numbers in Table S1. Raw reads count should not be used in Table S1. Normalised reads count like RPM (reads per million) should be used.
Lines 171-175: It is unclear what the biological significance is to compare miRNA expression between stress treatments.
Lines 191-196 & lines 581-583: Which miR169, miR395 and miR396? There are many miRNAs under the same family number in Table S1. Were the qPCR products sequenced to ensure the specificity of the primers? Same issue with the results of other miRNAs in this section.
Line 218: It is unclear how the targets were chosen. Did the authors identify the targets in literature or perform target prediction?
Line 274: Target gene expression can be repressed via other mechanisms. RACE experiment can validate the mRNA cleavage mode. qPCR does not provide enough supporting evidence.
Lines 328 - 331: Profile of only three modules does not provide enough supporting evidence for this statement.
Lines 398 - 402: Reference is needed for “real-world” biological significance.
Author Response
Dear Editor,
On behalf of the authorship team, I would like to take this opportunity to thank the two Reviewers for taking the time to comprehensively review our manuscript submitted for consideration for inclusion in the upcoming Special Issue of Plants. After careful consideration of each of the points raised by each Reviewer, we provide our responses below. I hope you find that we have adequately addressed the Reviewers’ concerns in our revised manuscript. Please note; for those points of concern raised by either Reviewer that we have not addressed in the revised version of our manuscript, we provide detailed justification as to why below.
Regards,
Andrew Eamens.
Reviewer #2 comments for plants – 450616
Comments and Suggestions for Authors
The study profiled the Arabidopsis microRNAome under drought, heat and salt stress. qPCR was used to analyse miRNA-target expression profile. The presentation of the results needs to be improved. Given that Arabidopsis microRNAs have been relatively well-documented in previous studies, the novelty and the highlights of this research need to be strengthened.
2.1. Line 83: Some good reference on utilising small RNAs for crop improvement. DOI: 10.1016/j.plantsci.2018.05.031; DOI: 10.1016/j.tplants.2016.07.006.
2.2. Line 87: Not really sure if this can be considered the first step, given that many studies have conducted sequencing analysis of Arabidopsis microRNAome under abiotic stress.
2.3. Lines 102-104: I don’t think there is enough evidence to support this statement. There is no molecular modification nor functional characterisation?
2.4. Lines 153-155: The change of one gene doesn’t really suggest changes of the whole global RNA expression.
2.5. Lines 531 & 563: Was total RNA treated with DNase? Complete removal of genomic DNA is critical for sRNA sequencing and qPCR to ensure accuracy.
2.6. Line 550: How many mismatches were allowed for miRNA annotation?
2.7. Line 553: miRNAs with low RPM (normally less than 10) should be omitted from fold-change analysis. See examples. DOI: 10.1186/s12870-014-0196-4; DOI: 10.1371/journal.pone.0142799; DOI: 10.1371/journal.pone.0139658
2.8. The fold-change of some miRNAs (e.g. miR168k) in table S1 should not be included.
2.9. Line 152 onwards: Figure 2 and the Results section should be revised after removing the miRNAs with low RPM.
2.10. Lines 162-170: None of the fold-change examples matches the numbers in Table S1. Raw reads count should not be used in Table S1. Normalised reads count like RPM (reads per million) should be used.
2.11. Lines 171-175: It is unclear what the biological significance is to compare miRNA expression between stress treatments.
2.12. Lines 191-196 & lines 581-583: Which miR169, miR395 and miR396? There are many miRNAs under the same family number in Table S1. Were the qPCR products sequenced to ensure the specificity of the primers? Same issue with the results of other miRNAs in this section.
2.13. Line 218: It is unclear how the targets were chosen. Did the authors identify the targets in literature or perform target prediction?
2.14. Line 274: Target gene expression can be repressed via other mechanisms. RACE experiment can validate the mRNA cleavage mode. qPCR does not provide enough supporting evidence.
2.15. Lines 328 - 331: Profile of only three modules does not provide enough supporting evidence for this statement.
2.16. Lines 398 - 402: Reference is needed for “real-world” biological significance.
As for Reviewer #1, the authorship team thank Reviewer #2 for their assessment of our submitted manuscript and for providing the above helpful comments for improvement of our submitted study.
2.1. As indicated by Reviewer #2, these two informative studies are now cited in the text of the revised manuscript (see line 83).
2.2. We have modified the wording of this sentence in order to take on-board this comment by Reviewer #2 (see lines 87-88).
2.3. The wording of the identified sentence has been changed to address this concern of Reviewer #2 (see lines 102 to 107).
2.4. We agree with this comment of Reviewer #2 and have therefore modified the wording of the associated sentences (please see lines 132 to 135 and lines 162 to 163 in the revised manuscript).
2.5. Please see lines 548 to 550 of the revised manuscript. We did not treat the total RNA with DNase prior to the total RNA being shipped to the commercial supplier, the AGRF (Melbourne Australia), for sequencing. We prefer to send untreated total RNA preparations to the AGRF where all subsequent preparatory steps, including DNase treatment, are performed by the company. We thank Reviewer #2 for identifying this oversight in our originally submitted manuscript: yes we routinely treat our total RNA extractions within DNase I prior to these preparations being subsequently used as template for either miRNA-specific cDNA synthesis (please see revised manuscript lines 572 to 573 for this correction) or standard cDNA synthesis (outlined in the originally submitted manuscript).
2.6. We again thank Reviewer #2 for raising this concern and we have added a sentence in the revised version of the manuscript that specifically states that only perfectly matched sequences were included at this stage of the analysis (please see lines 558 to 560 of the revised manuscript).
2.7. The authorship team are aware of the standard removal of miRNAs (or other sRNAs) classed as lowly abundant, that is; miRNAs with reads of less than 10 reads are removed. However, only miRNAs with reads of greater than 10 reads in at least one of the four generated libraries were listed in Table S1, and therefore included in the fold change analysis presented in Figure 2A; analyses that were performed on reads values determined on total library sizes (to determined RPM values).
2.8. The reads listed in Table S1 are raw reads whereas the read values used to determine the fold changes depicted in Figure 2A were first converted to RPM (total libraries sizes are presented at the bottom of the corresponding column in Table S1) to allow for their inclusion in these fold change calculations.
2.9. Please see points 2.7. and 2.8. above.
2.10. In the text of the manuscript, miRNA abundance is discussed as standard fold change. Yes these numbers differ from the fold change values presented in Table S1 as they are presented as Log2 fold changes in order to generate the data presented in Figure 2A. Further, while the raw reads are presented in Table S1, the total library size of each library is provided at the bottom of Table S1 in the corresponding columns to allow for the determination of RPM. Please also see points 2.7. and 2.8. above.
2.11. The authors again thank Reviewer #2 for identifying the requirement to expand the discussion of the observed findings. In order to address this concern of Reviewer #2, an explanatory sentence has been added to the end of this section of text (please see lines 193 to 196 of the revised manuscript).
2.12. Taking miR169 and heat stress treatment as an example; 11 out of the 12 members of the MIR169gene family detected here were determined to have elevated abundance post heat stress treatment. The mature sRNA sequence of each detected family member was aligned 5’ to 3’ and a consensus sequence was generated. The generated sequence was subsequently used to design a stem-loop primer for the reverse transcriptase reaction that could potentially hybridise with all detected miRNA family members. An explanatory sentence has been added to the revised manuscript (please see lines 606 to 609 in the revised manuscript). We can further confirm that all amplicons generated as part of this analysis were cloned and sequence to ensure that correct sequence was under assessment. However, we have not included the obtained data in this manuscript.
2.13. The miRNA target genes included in these analyses were all selected based on previously publish literature. Further for each miRNA under assessment, target genes were selected on the basis of previous demonstration that their expression was under the miRNA-directed mRNA cleavage mode of RNA silencing. This approach was selected to ensure that readily apparent miRNA / miRNA target gene abundance trends were obtained, that is; reduced miR169 abundance post heat stress treatment inversely correlates with elevated NFYA5 expression in heat stressed plants.
2.14. We have previously published two research articles in Nature Plants (PMID: 27246880) and the Journal of Proteome Research (PMID: 26387911) documenting miRNA-directed target transcript cleavage and miRNA-directed translational repression as distinct modes of target gene expression regulation in Arabidopsis. The findings reported in these two studies were used as part of our selection of miRNA target genes under expression regulation mediated via the target transcript cleavage mechanism of RNA silencing for quantification via the RT-qPCR analyses reported here. Combining this approach with that outlined above in 2.13., places RACE analysis out of the scope of this study.
2.15. We agree with Reviewer #2 comments on this sentence and have modified the sentence accordingly. Hope the modified sentence is a more accurate reflection of the degree of results presented in manuscript Figure 5 (please see lines 348 to 351 of the revised manuscript).
2.16. We have considered this concern of Reviewer #2 and accordingly have modified the identified sentence (see line 422 of the revised manuscript).
Round 2
Reviewer 1 Report
The current version of manuscript could be accepted.
Reviewer 2 Report
Accept